# Flame Spraying of Aluminum Coatings Reinforced with Particles of Carbonaceous Materials as an Alternative for Laser Cladding Technologies

**DOI:** 10.3390/ma12213467

**Published:** 2019-10-23

**Authors:** Artur Czupryński

**Affiliations:** Welding Department, Faculty of Mechanical Engineering, Silesian University of Technology, ul. Konarskiego 18A, 44-100 Gliwice, Poland; Artur.Czuprynski@polsl.pl

**Keywords:** flame powder spray process, coating, aluminum, carbon nanotubes, carburite, abrasive wear resistance, erosion wear resistance

## Abstract

The article presents results of the preliminary research of mechanical properties of flame-sprayed aluminum coatings reinforced with carbon materials made on the construction steel S235J0 substrate. For reinforcement the following carbon materials were used: carbon nanotubes Nanocyl NC 7000 (0.5 wt.% and 1 wt.%) and carburite (0.5 wt.%). The properties evaluation was made using metallographic macroscope and microscope, chemical composition, microhardness, abrasion and erosion resistance studies. The obtained results were compared with aluminum powder coatings (EN AW 1000 series). It was proved that the flame spraying of aluminum coatings reinforced with particles of carbonaceous materials can be an effective alternative for laser cladding technology. The preliminary test results will be successively extended by further experiments to contribute in the near future to develop innovative technologies, that can be implemented in the automotive industry for production of components with high strength, wear resistance, good thermal conductivity and low density, such as brake shoes, cylinder liners, piston rings and gears.

## 1. Introduction

Modern civilization expects from material engineering scientists to produce lightweight and durable materials that meet the high strength and quality requirements set for innovative constructions made by the automotive and aerospace industries. Under certain structural load conditions, the increase of strength and stiffness of the materials contributes to reducing construction dimensions and consequently also the mass. Because the global oil resources are constantly declining, and renewable energy sources are not effective enough yet, use of lightweight and durable materials becomes a necessity. This type of materials is highly desirable in car, aircraft and space vehicle production because its use has many benefits, such as lower fuel consumption, higher capacity and speed. Insufficient strength and stiffness of the constructions made of metals and alloys led to the development of metal-matrix composites (MMC). The composites of this type achieve high strength and ductility thanks to the metallic matrix, while the stiffness is provided by the reinforcement, which consists of particles-perchance fibers-metallic or ceramic with high stiffness. The microstructure of this materials consists of soft matrix and hard phases which provides increase in abrasion resistance also at high temperatures. Metal-matrix composites can be designed to have specific properties, such as low thermal expansion coefficient and high thermal conductivity so that these materials are suitable for use in applications for installation of electronic microcircuits. Metal-matrix composite materials are widely used in car and air applications nowadays [1,2,3].

In the 1970s, technologies for producing high-strength carbon fibers were developed. They began to be used for the preparation of advanced composites used for producing rocket engine nozzles, projectile cores, thermal shields, isolators and thermal radiators. Since 1970, carbon fibre reinforced composites have been widely used in the production of aircraft brakes, space constructions, military and commercial airplanes, lithium-ion batteries and sports equipment. Research in the field of carbon materials has been revolutionized by the discovery of carbon nanotubes (CNT) by Sumio Iijima in 1991. The carbon nanotubes (CNTs) have unique mechanical properties compared to carbon fibers, e.g. stiffness to 1000 GPa, strength of 100 GPa and thermal conductivity to 6000 W/(m·K) [4,5]. In recent years, a number of studies have been carried out using CNT carbon nanotubes as reinforcement of various materials: polymers, ceramics and metals, with the majority of research involving polymer composites [6,7], ceramic composites in second place [8,9], and only recently have been published several papers on composites with metallic matrix reinforced with carbon nanotubes (CNT) [10,11]. This is quite surprising considering the fact that most construction materials used in the contemporary world are metals. Publications on this topic concern various aspects such as fabrication [12,13,14,15], microstructure [16,17], modelling of mechanical properties and the chemical interaction between carbon nanotubes (CNTs) and metals [4,18,19,20,21]

Nanotechnology had a strong influence on the direction of research in the field of surface engineering and related production technology of surface layers and coatings [22,23]. Nowadays, it is possible to use welding methods for producing not only conventional tribological coatings with specific frictional characteristics (high or low coefficient of friction) and resistance to wear, erosion or corrosion but also for producing coatings with unprecedented properties, often intended for special applications and working in difficult conditions, e.g., nanocomposite coatings with high hardness and high resistance to dynamic loads, coatings with frictional characteristics that adapt to changing operating conditions (temperature, humidity), thermal barrier coatings or biocompatible coatings [24,25]. Often, high-quality nanostructure coatings are used on parts of car engines made of aluminum alloys, on copper alloys intended for propellers of vessels, or on heat-resistant intermetals. In surface engineering technology, the implementation of this type of coating is possible by thermal spraying, where the applied metallic layer is bonded to the substrate adhesively or mechanically without melting the base material [24]. The main advantage of thermal spraying technology is a minimal thermal influence on the sprayed materials. Even in the case of laser cladding technologies characterized by the lowest heat input of all the cladding technologies, the substrate material is always partially melted, as well as the additional material, usually in a form of metallic or composite powder. The carbon nanotubes, due to the small dimensions, have very low heat capacity. Additionally, they have high absorption of laser radiation. For this reason, the introduction of carbon nanotubes into the melt pool during laser cladding is basically impossible, because overheating and decomposition of nanotubes [26,27,28,29,30,31,32,33].

Pioneers in the field of thermal spraying processes for composite coatings of aluminum-carbon nanotubes (CNT) were a research group from Florida International University, who successfully deposited carbon nanotubes in the Al-Si matrix in the powder plasma spraying process [34].

S. R. Bakshi and others [10] made multi-layer nanocomposite coatings of aluminum-carbon nanotubes (CNT) in the cold gas spraying process. In order to obtain a good dispersion of carbon nanotubes in Al-Si microparticle eutectic powders, spray drying was used. Spray-dried powders containing 5 wt.% carbon nanotubes (CNTs) were mixed with pure aluminum powder to obtain total nominal carbon nanotube (CNT) compositions in the coating material of 0.5 wt.% and 1 wt.%. As a result of cold spraying, coatings with a thickness of 500 μm were obtained in which the carbon nanotubes were evenly distributed in the matrix. The carbon nanotubes were of shorter length because during the deposition process they fractured as a result of impact and shear between the Al-Si particles and the aluminum matrix.

A. K. Keshri and others [11] compared impact on carbon nanotubes (CNTs) of various heat sources used during thermal spray processes—plasma spraying (PS), high-velocity oxy fuel spraying (HVOF), cold spraying (CS) and plasma spraying of liquid precursor (PSLP). Carbon nanotubes (CNTs) have been successfully preserved as reinforcements in composite metal and ceramic coatings in all thermal spray processes with the exception of PSLP.

There is no data in the literature regarding tribological properties of powder flame-sprayed (PFS) aluminum coatings reinforced with carbon nanotubes (CNT). The purpose of this article is to present the state of knowledge in this area of research and present the possibility of using powder flame spray technology (PFS) for the production of composite coatings with a metallic matrix reinforced with carbon nanotubes (CNT).

## 2. Materials and Methods

### 2.1. Aim of Study

The conducted studies were aimed at comparing the structure, chemical composition, hardness and resistance to abrasive and erosive wear of powder aluminum flame-sprayed coatings reinforced with Nanocyl NC 7000 carbon nanotubes in amount of 0.5 wt.%, 1 wt.% and carburite in an amount of 0.5 wt.% with a reference coating made of aluminum powder EN AW 1000 series (Metallisation Ltd., West Midlands, UK) on non-alloy S235J0 steel. Carburite as aluminum matrix reinforcement was used in order to compare tribological properties of this composite coating with coating reinforced with CNTs with equal weight participation of carbon material. The scope of research included:Preparation of material for spraying;Selection process parameters for each of the coating based on preliminary technological tests;Coating manufacturing;Examining the structure and tribological properties of aluminum coatings reinforced with carbon nanotubes and carburite;Comparison of obtained samples with coatings made of aluminum powder without the addition of carbonaceous materials.

### 2.2. Materials, Devices and Spraying Parameters

The additional material for flame-spraying was obtained by mixing in the ball mill appropriate proportions of aluminum powder (EN AW 1000 series) with carbon nanotubes and aluminum powder in the form of filter dust carburite (Table 1). Carbon nanotubes that were used in the test are commercially available multi-walled carbon nanotubes MWCNTs, produced in the Catalytic Chemical Vapor Deposition (CCVD) process, NANOCYLTM NC7000 (Belgium Nanocyl SA, Sambreville, Belgium) (Table 2).

The subsonic flame spraying process was carried out cold in accordance with the standard EN 13507:2018 [35] on workstation, equipped with hand-guided modern oxyacetylene system (CastoDyn DS 8000 (Messer Eutectic Castolin, Gliwice, Poland). Final surface preparation was done by shot blasting sheets prior to spraying with sharp-edged cast iron of 0.5–1.5 mm shot grain size in accordance with standard ISO 2063-1:2017 [36]. Final surface roughness of the steel substrate after shot blasting was R_a_ = 12 μm, R_z_ = 85 μm. Before the spraying process, steel plates with dimensions of 150 × 150 × 5 mm^3^ were preheated with a gas burner up to a temperature of 40 °C (the temperature of preheating was measured using pyrometer). The standard modular nozzles regulating the flame outlet SSM 40 (Messer Eutectic Castolin, Gliwice, Poland) and the neutral flame (ratio O_2_/C_2_H_2_ = 1,2) were used. This allowed to obtain the proper spray jet [37,38]. The flame jet burner was guided in a horizontal position covering the whole surface of the sheet. During the process the spraying direction was changed several times by 90°, until obtained thickness of coating was about 1,0 mm. Distance between the torch and the sprayed surface was 200 mm. The parameters and flame type were constant for each test (Figure 1).

The criterion for visual assessment of the powder coatings’ quality, was to make the surface layers characterized by the appropriate thickness, good adhesion to the substrate, low porosity, continuity and uniformity of obtained coatings [39]. Optimal parameters of flame-spraying of aluminum, aluminum with carbon nanotube reinforcement and aluminum with filter dust carburite reinforcement coatings have been determined on the basis of preliminary technological tests (Table 3). The view of representative samples with flame-sprayed coatings on the aluminum matrix are shown in Figure 2.

### 2.3. Visual and Metallographic Examination of Coatings

In each case, the entire surface of the sample was subjected to visual tests to assess the quality and identify any imperfections in the form of cracks, discontinuities, unevenness, porosity or lack of coating adhesion. Macro and microscopic examinations were performed on Olypmus GX 71 optical microscope (Olympus Corporation, Tokyo, Japan). The observations were made on the cross-section of metallographic samples cut from the centre of element. Samples were polished and etched in Aqua Regia. Selected areas of flame-sprayed coatings (aluminum and aluminum with addition of carbon materials) have been subjected to chemical composition analysis on JEOL 5800LV SEM scanning microscope and also EDX (Jeol Ltd., Tokyo, Japan).

### 2.4. Hardness Measurements of Coatings

The coating-hardness measurement was made with the Vickers method using Microhardness Tester 401MVD™ (Wilson Instruments An Instron Company, Norwood, MA, USA). The examinations were carried out in conformity to ISO 6507-1:2018 standard [40]. The load applied during the hardness measurement was 0.98 N. The hardness measurement was made at the polished cross-section of the samples with flame-sprayed coatings. Ten hardness measuring points were made on the cross-section each sprayed coating.

### 2.5. Erosive Wear Resistance of Coatings

The erosive wear tests of flame-sprayed coatings were carried out in accordance with ASTM G76-95 [41], as shown in Figure 3. Aluminum oxide powder (Al_2_O_3_) with particle diameter of 71 µm was used as the erodent material. Particle velocity was set at 70 ± 2 m/s, the erodent expenditure was 2.0 ± 0.5 g/min and the nozzle distance from the sample surface was 10 mm. The tests were carried out at 90° and 30° erodent impact angle. The average weight loss was determined based on three tests. The erosion rate was determined according to the Equation (1),
erosion rate [g/min] = mass loss of sample [g]:exposure time [min](1)
However, the erosive wear resistance using Equation (2):erosive wear resistance [0.001mm^3^/g] = volume loss of the sample [mm^3^]: total mass of the erodent used in the test [g](2)

### 2.6. Abrasive Wear Resistance of Coatings

Metal-mineral wear resistance tests of aluminum matrix coatings were provided in accordance with ASTM G65-00, Procedure E [42]. During the test, the rubber-wheel made 1000 revolutions and the abrasive flow rate of A.F.S. Testing Sand 50–70 was 335 g/min. The force applied pressing the test coupon against the wheel was TL = 130 N (test load-TL). After the abrasive wear resistance test, the test specimen was weighed at weight sensitivity 0.0001 [g]. Converting mass loss to volume loss was as follows:volume loss [mm^3^] = mass loss [g]:density [g/cm^3^] x 1000(3)

The tests were carried out on abrasion tester shown in Figure 4.

## 3. Results

### 3.1. Metallographic Test Results

The structure of each tested flame-sprayed coating cross-section is presented in Figure 5. The SEM structures of tested aluminum matrix coatings with chemical composition are presented in Figure 6, Figure 7, Figure 8 and Figure 9.

### 3.2. Hardness Measurements

The hardness measurements on flame-sprayed aluminum and aluminum matrix reinforced with carbon material coatings, were carried out on the surface at 5 points along one measuring line (Table 4) and on cross-section of the samples (Figure 10 and Figure 11), according to the scheme showed on Figure 10.

### 3.3. Tests Results of the Coatings Erosive Wear Resistance

The relative erosive wear resistance test results of the flame-sprayed aluminum, aluminum with carbon nanotube reinforcement and aluminum with filter dust carburite reinforcement coatings are presented in Table 5 and Figure 12.

### 3.4. Tests Results of the Coatings’ Wear Resistance

The wear resistance test results of the flame-sprayed aluminum, aluminum with carbon nanotube reinforcement and aluminum with filter dust carburite reinforcement coatings are presented in Table 6 and Figure 13. The metal-mineral type wear resistance of the flame-sprayed aluminum with carbon nanotubes and aluminum with carburite coatings were compared to the flame-sprayed pure aluminum coating.

## 4. Discussion

Visual and metallographic tests of the flame-sprayed aluminum and aluminum with carbon material reinforcement (0.5 wt.% and 1 wt.% of carbon nanotubes Nanocyl NC 7000 and 0.5 wt.% of carburite) have shown that by using proper parameters of the process it is possible to receive coatings with acceptable quality level, characterized by proper adhesion to the substrate, lack of delamination and even thickness over the entire surface. The outer surface of the coatings was characterized by a slight roughness, lack of porosity and cracks (Figure 2). During the flame-spraying process, carbon material particles added to aluminum powder did not oxidize completely in the oxyacetylene flame. Carbon nanotubes (melting point 4526 °C [43]) and carburite (melting point 3550 °C [44]) in the oxyacetylene flame has formed with aluminum Al-Cx type agglomerates, which due to the large volume and lower heat source temperature than other thermal spraying methods (oxyacetylene flame temperature 3160 °C [45]) migrated in large quantities to the coatings. Partially melted and partially only plasticized in a gas flame, Al-Cx agglomerates collided with the substrate at high speed, (Figure 1b) and in this way formed a fine-grained coating structure. Powder flame spraying process (PFS) in comparison with, for example, plasma spraying, increases the probability of stopping carburite and carbon nanotubes (CNT) in flame sprayed composite coating with aluminum matrix. Presence of carbon materials in aluminum powder flame-sprayed coatings is initially confirmed by metallographic microscopic tests, which revealed areas carburite and carbon nanotubes on specimens (Figure 5, Al + 0.5% CNT). Presence of carbon materials can be observed on the entire cross-section of the coating, also at the outer surface. Inside the Al-Cx composite coatings, no cracks were found, only the presence of individual cavities. The tests made using scanning electron microscope have shown presence of some areas consisting small carbon materials inclusions. These were observed in the all-aluminum coatings with carbon material reinforcement. For the coating with 0.5 wt.% CNT, inclusion areas consisted of 33.05 wt.% C; for coating with 1 wt.% CNT, the carbon content was lower in tested area (20.25 wt.% C), while for the coating with 0.5 wt.% of carburite, carbon content was almost two times higher than in the coating with same content of CNTs and amounted to 59.76 wt.% C. In aluminum coating without carbon material addition, carbon and oxygen were found, (Figure 6). A small amount of carbon in the aluminum coating may be caused by the ease of thermal decomposition of acetylene in the gas flame and the physicochemical properties of unsaturated hydrocarbons [46]. Acetylene is dissociated into active carbon atoms (acetylene black, characterized by high purity) and hydrogen molecule. The oxygen content in the aluminum coating is the result of oxidation of the aluminum particles in the gas flame and the atmosphere. The addition of carbon materials to the aluminum powder causes the carbon to bind oxygen as a strong deoxidizer; that is why its presence was not found in composite coatings with aluminum matrix and carbon material (carburite and carbon nanotubes) reinforcement. These results should still be confirmed using more advanced research methods, e.g. XRD X-ray diffraction or Raman spectroscopy. These studies will be completed and presented in another publication.

The hardness measurements of tested coatings were proceeded using standard ISO 6507 [40]. The measurements were done both on the external surface and the cross-section of the sprayed coatings. These tests showed that using addition of 0.5 wt.% and 1 wt.% carbon nanotubes to aluminum coating (34.1 HV 0.1) caused an increase in its hardness of 8.2 HV 0.1 for the 0.5 wt.% of carbon nanotube reinforcement and by 9.5 HV 0.1 for 1 wt.% of carbon nanotube reinforcement. Addition of carburite to aluminum had not significant influence on the coating hardness (Figure 11).

Erosive wear resistance test results have shown that the addition of carbon materials to aluminum powder does not increase the erosive wear resistance of flame-sprayed coatings. During these tests, the aluminum coatings with carbon nanotubes had worn out by erosion with large and small angles of erodent incidence more than aluminum coatings with carburite and much more than aluminum coatings without carbon materials reinforcement. It was observed that for all tested coatings erosive wear resistance was better during using smaller angle of erodent incidence (Table 4).

The best metal-mineral type wear resistance had the aluminum coating with carburite. The wear resistance of this coating was 19% higher than pure aluminum coating. The aluminum coatings with addition of 0.5 wt.% and 1 wt.% of carbon nanotubes in comparison to pure aluminum coating had better relative wear resistance by 10% and 11% (Table 5). The cause of decreasing wear using aluminum coatings with carbon material addition was increased glide of ceramic abrasive particles on metal.

Based on the conducted study and the obtained results, it can be concluded that it is possible to introduce carbon particles in the form of carbon nanotubes (CNT) and also carburite into the aluminum matrix by means of flame spraying method. The flame spraying is an effective and cheaper alternative to the technology of laser surface treatment of metals. Properly selected parameters of the flame spraying process allow to preserve the properties of particles of carbon materials, their even distribution in the coating, proper bonding with the matrix and prevent the effects of their thermal degradation. The produced composite is characterized by a low friction coefficient. The tribological characteristics of produced test coating of aluminum reinforced by carbon particles in the form of carbon nanotubes and carburite show that the coatings can be classified as sliding materials. Additionally, the coatings are characterized by high wear resistance. The obtained result should be considered as a preliminary information on a new group of materials, which can find application in the automotive industry. They are the basis for the design and optimization of friction materials operated at elevated temperatures (e.g. pistons, engine blocks), systems subject to intensive wear (e.g. brake discs, cylinders), as well as in propulsion systems (e.g. bearings), providing low friction coefficient and also high ability for absorption of vibration. Further research should be focused on the investigation of the effect of doping the aluminum matrix with carbon nanotubes (CNT) on the wear mechanisms, change of microstructure of the counter-specimen and also tests which will allow to determine the tribological characteristics of the materials at elevated temperatures.

## 5. Conclusions

The analysis carried out comparing properties of flame-sprayed EN AW 1000 aluminum coatings and aluminum matrix coatings with carbon materials reinforcement (0.5 wt.% and 1 wt.% of Nanocyl NC 7000 carbon nanotubes and 0.5 wt.% of carburite) resulted in the following conclusions:Producing aluminum matrix coatings with carbon materials reinforcement with high quality is possible using flame spraying technology.In the aluminum with carburite reinforcement flame-sprayed coating structure, areas were observed with a share of carbon above 61 wt.%. In the aluminum coating with 1 of carbon nanotubes, similar areas were observed with a share of carbon about 33 wt.%.The addition of carbon nanotubes to aluminum powder resulted in increasing the hardness of flame-sprayed coatings by about 10 HV 0.1.The carbon materials reinforced aluminum flame-sprayed coatings have lower erosive wear resistance than pure aluminum coatings with large and small angles of erodent incidence.The metal-mineral type wear resistance of flame-sprayed aluminum coatings reinforced with carbon nanotubes or carburite is 10% to 20% higher in comparison to pure aluminum coating.

## Figures and Tables

**Figure 1 materials-12-03467-f001:**
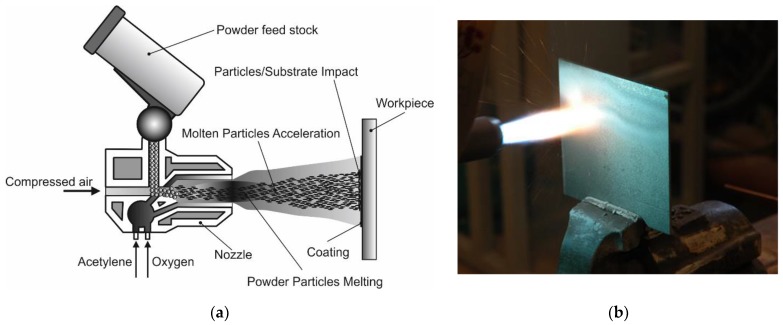
Flame-spraying process: (**a**) a diagram of handheld flame jet burner; (**b**) a photo from trials of flame-spraying aluminum coatings with CastoDyn DS 8000 burner.

**Figure 2 materials-12-03467-f002:**
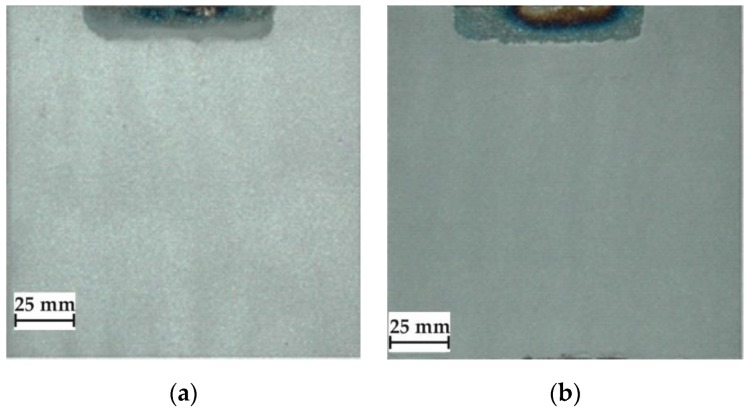
View of test samples with flame-sprayed coatings on the aluminum matrix: (**a**) aluminum powder of the EN AW 1000 series; (**b**) aluminum powder EN AW 1000 series with addition of 0.5 wt.% carbon nanotubes (Nanocyl NC 7000); **(c**) aluminum powder of the EN AW 1000 series with the addition of 1 wt.% carbon nanotubes (Nanocyl NC 7000); (**d**) aluminum powder of the EN AW 1000 series with the addition of 1 wt.% carburite.

**Figure 3 materials-12-03467-f003:**
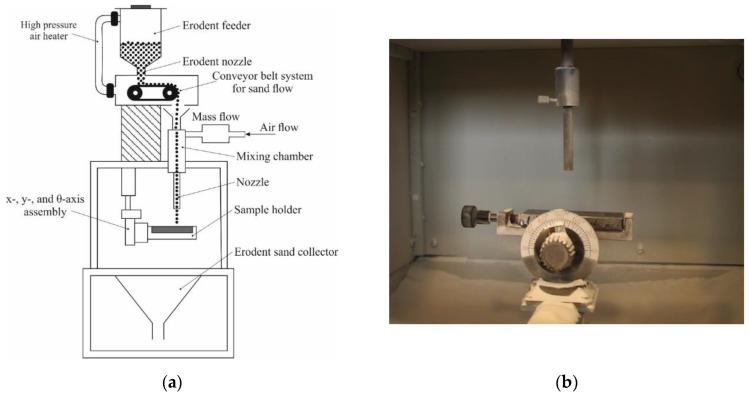
Erosion resistance testing according to ASTM G 76-95: (**a**) a schematic view, (**b**) the interior view of the erosion measuring chamber.

**Figure 4 materials-12-03467-f004:**
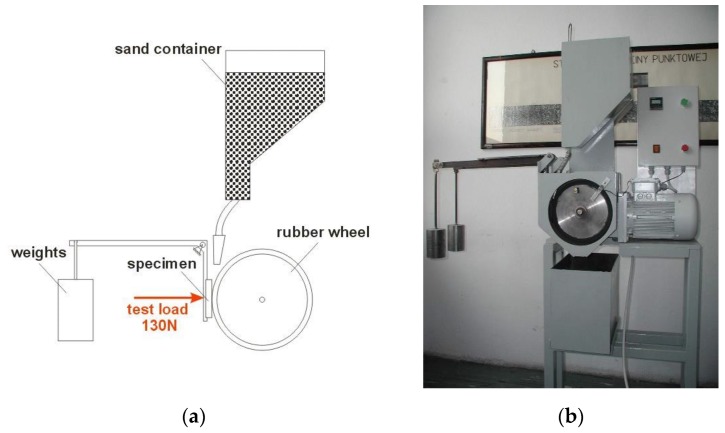
Metal-mineral wear resistance test stand–according to ASTM G65–00, Procedure E standard (**a**) a schematic view, (**b**) picture of the device used.

**Figure 5 materials-12-03467-f005:**
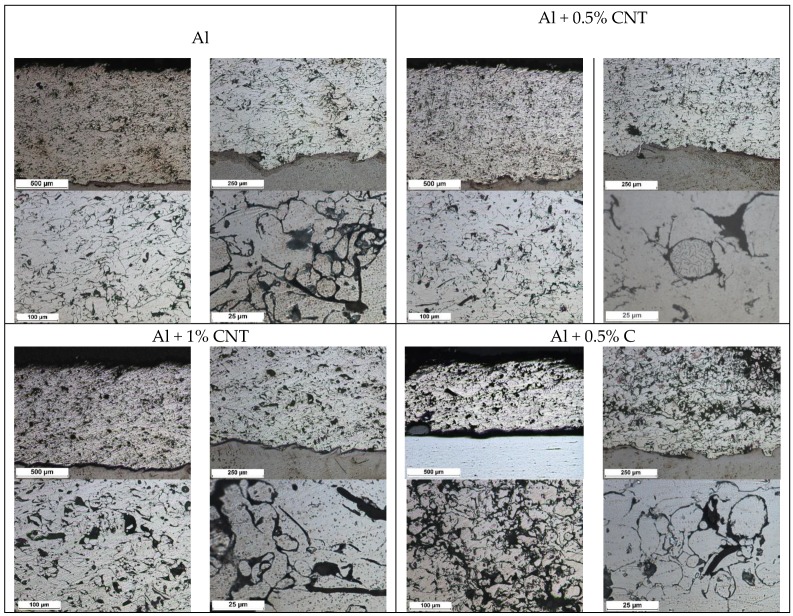
The macro and microstructure of the flame-sprayed pure aluminum and aluminum matrix with carbon nanotubes and carburite reinforcement coatings; etching: HCl + HNO_3._

**Figure 6 materials-12-03467-f006:**
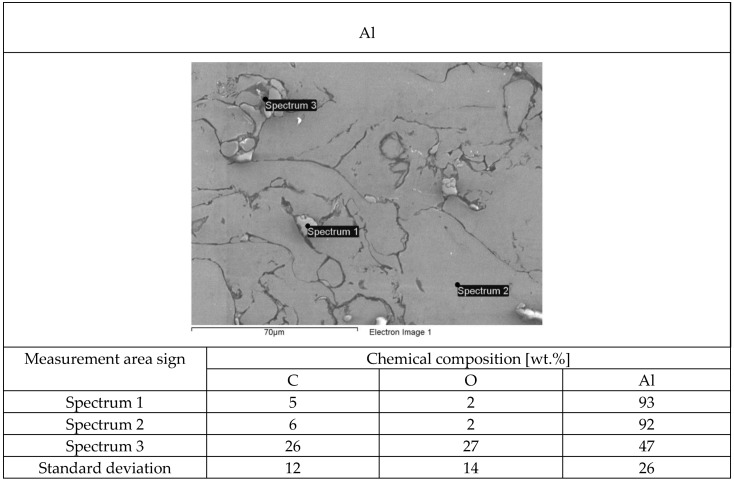
The structure of EN AW 1000 aluminum powder flame-sprayed coating with marked chemical composition tested areas on SEM.

**Figure 7 materials-12-03467-f007:**
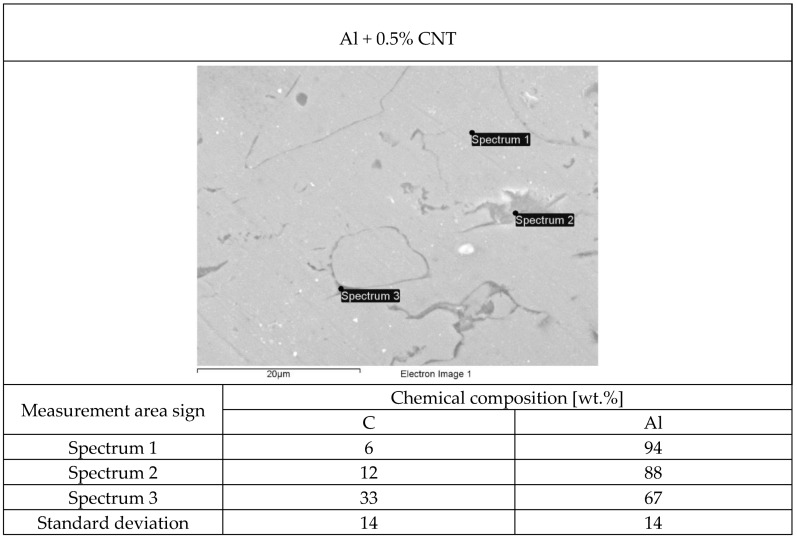
The structure of aluminum powder with 0.5 wt.% Nanocyl NC 7000 carbon nanotubes flame-sprayed coating with marked chemical composition tested areas on SEM.

**Figure 8 materials-12-03467-f008:**
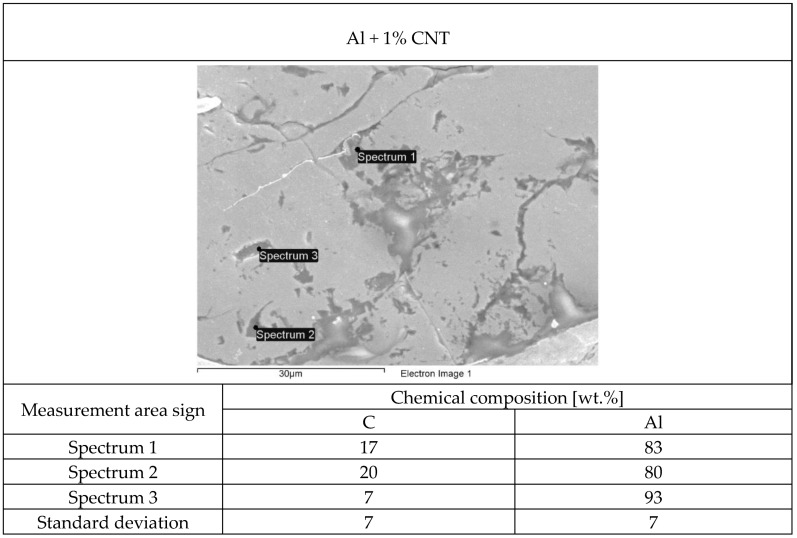
The structure of aluminum powder with 1 wt.% Nanocyl NC 7000 carbon nanotubes flame-sprayed coating with marked chemical composition tested areas on SEM.

**Figure 9 materials-12-03467-f009:**
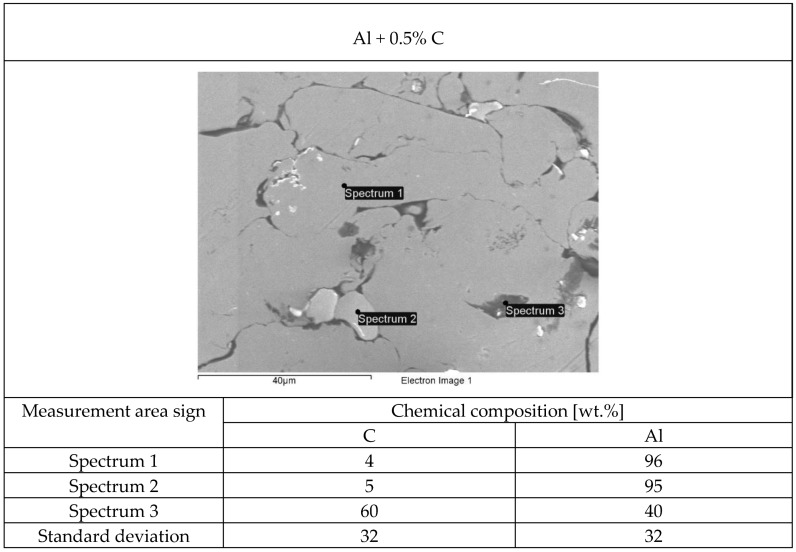
The structure of aluminum powder with 0.5 wt.% carburite carbon nanotubes flame-sprayed coating with marked chemical composition tested areas on SEM.

**Figure 10 materials-12-03467-f010:**
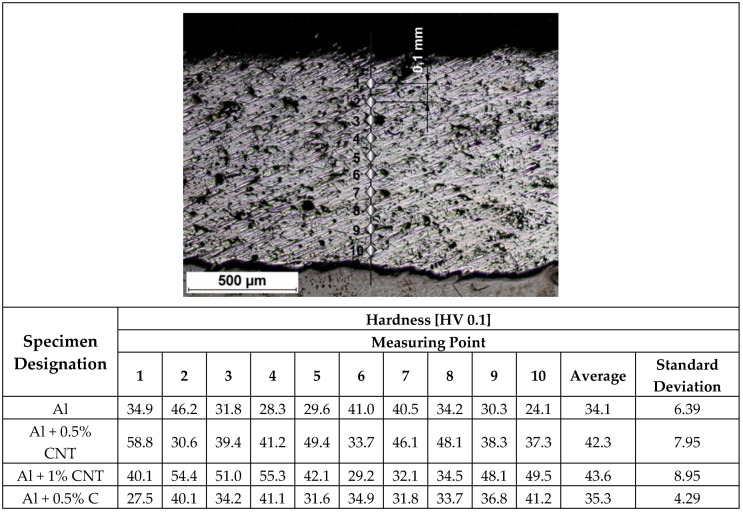
Hardness measurements scheme and HV 0.1 results of the flame-sprayed aluminum and aluminum reinforced with carbon materials coatings.

**Figure 11 materials-12-03467-f011:**
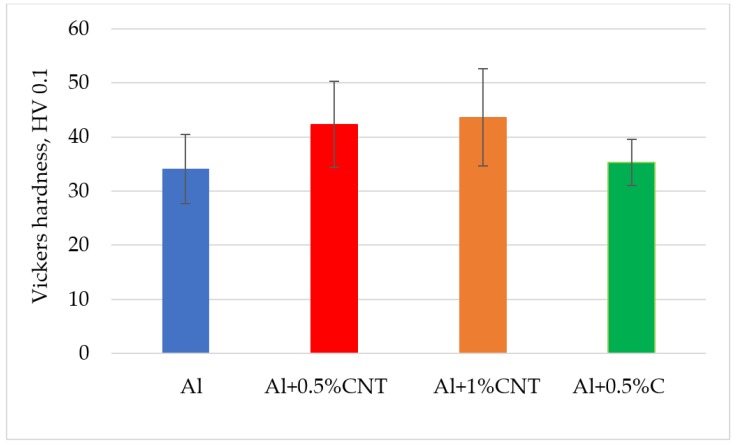
Comparison of average cross-sectional hardness and standard deviation for each of the investigated coatings.

**Figure 12 materials-12-03467-f012:**
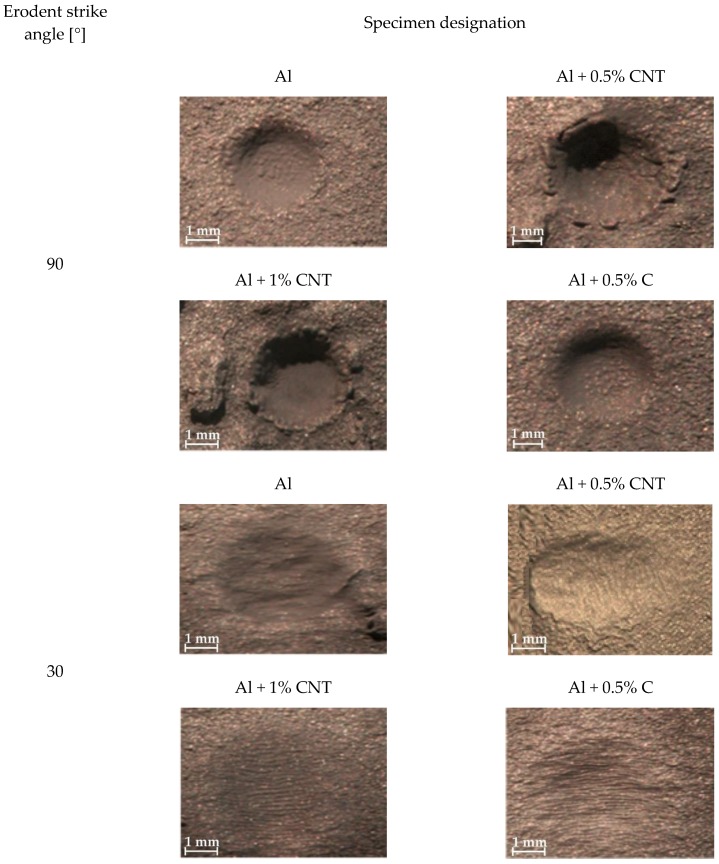
The surfaces of flame-sprayed aluminum and aluminum matrix reinforced with carbon material coatings after erosive wear resistance tests; comparison the erosion effect on samples surfaces for each tested angle of erodent particles incidence.

**Figure 13 materials-12-03467-f013:**
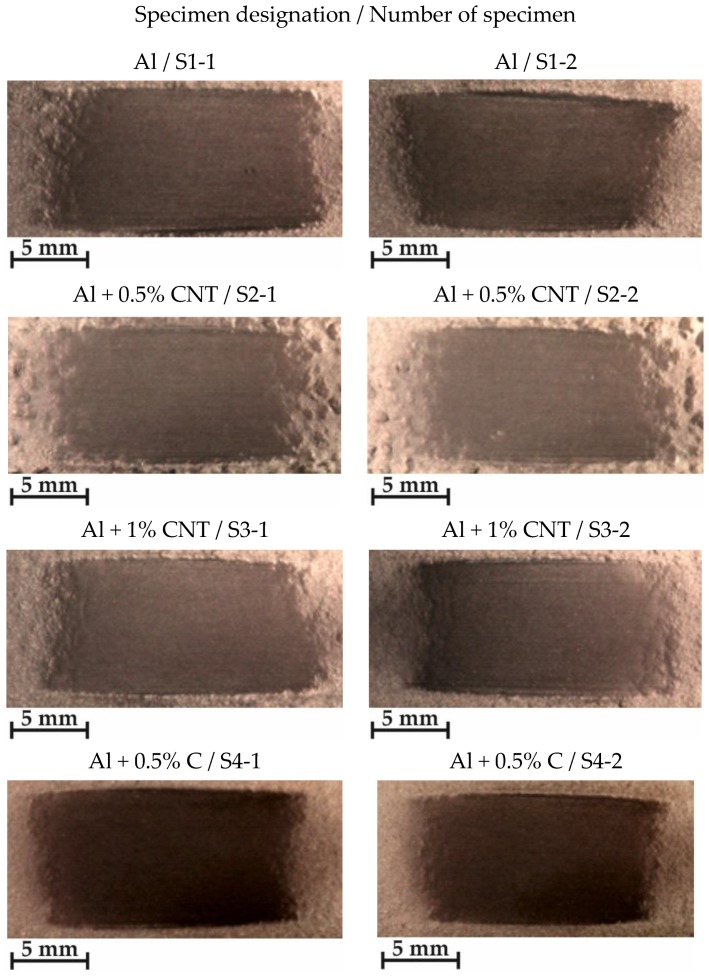
The surfaces of flame-sprayed aluminum and aluminum with carbon material coatings after wear resistance metal-mineral tests.

**Table 1 materials-12-03467-t001:** Specification of aluminum powder EN AW 1000 series.

Specification	Unit	Guaranteed Parameters	Obtained Results
Aluminum content (Al)	%	min 99.7	99.7
Iron content (Fe)	%	max 0.2	0.2
Silicon content (Si)	%	max 0.12	0.12
Copper content (Cu)	%	max 0.004	0.004
Moisture	%	max 0.1	0.1
Bulk density	g/dm^3^	min 1000	1050
Granulation above 0.045 mm	%	85.0−100.0	98.0
Granulation above 0.1 mm	%	5.0−30.0	15.3
Granulation above 0.16 mm	%	max 5.0	0.0

**Table 2 materials-12-03467-t002:** Structure and specification of Nanocyl NC 7000 carbon nanotubes and of filter dust carburite.

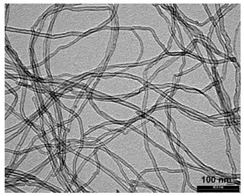 Structure of Nanocyl NC 7000 carbon nanotubes			
**Properties**	**Unit**	**Value**	**Measurement Method**
Average diameter	Nanometer	9.5	TEM
Average length	Micrometer	1.5	TEM
Carbon purity	%	90	TGA
Metal oxide	%	10	TGA
Amorphous carbon	-	^1)^	HRTEM
Surface area	m^2^/g	250–300	BET
Note: ^1)^ pyrolytically deposited carbon on the surface of the NC 7000.			
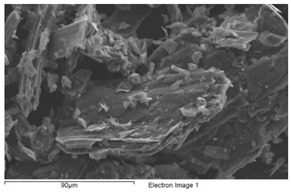 Structure of filter dust carburite			
**Properties**	**Unit**	**Value**	
Fraction size	Millimeter	>0 up to 1	
Dust content	%	5	
Moisture content	%	1	
Carburite content	%	94	
Granulation [mm]	%	>1 mm up to 10%; <0.06 mm up to 70%	

**Table 3 materials-12-03467-t003:** Parameters of flame-sprayed aluminum and aluminum coatings reinforced with carbon materials using CastoDyn DS 8000 torch.

Sample Number	Type of Powder	Oxygen Pressure [bar]	Acetylene Pressure [bar]	Assist. Gas (Compressed Air) Pressure [bar]	Number of the Orifice for the Powder	Mass of Used Powder [g]	Powder Yield [%]
1	Al	4	0.7	3	2	93.5	60.3
2	Al + 0.5% CNT ^1)^	4	0.7	3	2	97.0	56.2
3	Al + 1% CNT	4	0.7	3	2	100.8	57.0
4	Al + 0.5% C ^2)^	4	0.7	3	2	99.7	59.4

Note: ^1)^ CNT—carbon nanotubes Nanocyl NC 7000 wt.%; ^2)^ C—carburite wt.%.

**Table 4 materials-12-03467-t004:** Surface hardness results of the flame-sprayed aluminum and aluminum reinforced with carbon material coatings.

Specimen Designation	Hardness [HV 0.1]						
	Measuring Point						
	1	2	3	4	5	Average	Standard Deviation
Al	35.2	32.5	34.8	37.2	35.1	34.96	1.67
Al + 0.5% CNT	59.2	62.8	56.5	54.2	55.5	57.64	3.42
Al + 1% CNT	41.2	43.2	37.5	39.2	38.7	39.96	2.25
Al + 0.5% C	30.2	27.3	28.2	27.0	28.0	28.14	1.25

**Table 5 materials-12-03467-t005:** Summary of results obtained during the erosion wear test ASTM G76−95 [41].

Erodent Strike Angle [°]	Specimen Designation	Mass Loss [g]	Volume Loss [mm^3^]	Erosion Rate [g/min]	Resistance to Erosion as per ASTM G76 [0.001mm^3^/g]
90	Al	0.0054	1.985	0.00068	0.12255
	Al + 0.5% CNT	0.0117	4.301	0.00146	0.26552
	Al + 1% CNT	0.0071	2.610	0.00089	0.16113
	Al + 0.5% C	0.0064	2.353	0.00080	0.14524
30	Al	0.0036	1.324	0.00045	0.08170
	Al + 0.5% CNT	0.0066	2.426	0.00083	0.14978
	Al + 1% CNT	0.0045	1.654	0.00056	0.10212
	Al + 0.5% C	0.0039	1.434	0.00049	0.08851

Notes: density of aluminum spray coating 2.72 [g/cm^3^], mass of erodent used 16.2 [g], test time 8 [min].

**Table 6 materials-12-03467-t006:** Summary of results obtained during the abrasive wear test ASTM G65 [42].

Specimen Designation	Number of Specimen	Weight Before Test [g]	Weight after Test [g]	Mass Loss [g]	Average Mass Loss [g]	Average Volume Loss [mm^3^]	Relative ^1)^ Abrasion Resistance
Al	S1-1	43.9675	43.8413	0.1262	0.1418	52.1324	1.00
	S1-2	42.3855	42.2281	0.1574			
Al + 0.5% CNT	S2-1	56.5170	56.3924	0.1246	0.1286	47.26103	1.10
	S2-2	53.8604	53.7279	0.1325			
Al + 1% CNT	S3-1	56.9638	56.8322	0.1316	0.1279	47.0221	1.11
	S3-2	57.4587	57.3345	0.1242			
Al + 0.5% C	S4-1	59.4423	59.3199	0.1224	0.1190	43.7500	1.19
	S4-2	61.1152	60.9996	0.1156			

Notes: density of aluminum spray coating 2.72 [g/cm^3^]; ^1)^ relative to sprayed coatings of the aluminum without carbon materials.

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
