# Peer review of "Flame Spraying of Aluminum Coatings Reinforced with Particles of Carbonaceous Materials as an Alternative for Laser Cladding Technologies"

_materials, 2019, doi:10.3390/ma12213467_

Round 1

Reviewer 1 Report

The present paper present the state of knowledge in the area of MMC materials, namely metallic matrix reinforced with carbon nanotubes and investigated the using of powder flame spray technology for the production of the composite coatings.

Introduction is concise, systematized in order to guide the reader towards the objectives of the work.

I present some suggestion to be improved:

Line 49: “Studies have shown that carbon nanotubes…” - what studies? It is better present some references.

Line 56: “Publications on this topic concern various aspects such as …” – If this sentence is justified with examples, it promotes the likelihood of citing, e.g. several references related with fabrication, others with microstructure, modelling, etc.

Line 121: “Table 3” – No, Table 1.

Lines 134/135: Table 2 and Table 3 – In my opinion is better put these images together in a figure.

Line 138: “(usunalem tutaj and)” - Please correct

Line 147: Figure 2 – suggestion insert scale on the images.

Line 147: “Vickers HV0.1” – correspond to 0.98N. It is better insert the load.

Line 159: “SEM scanning microscope” and also EDX analysis.

Line 163: the cross-sections are polished before the indentations? The hardness values are very influenced by the porosity.

Figures 6, 7 and 8: The chemical composition has a higher standard deviation (higher than 6%) is not suitable to insert the chemical composition of each spectrum with two decimal digits.

Table 5, 6 and Table 7, and Figures 10 - 13: using the name of sample 1, sample 2, sample 3 and sample 4 in each row is repetitive and makes the information denser and harder to understand.

Line 206: please correct the vertical axis legend (with point): Vicker hardness, HV0.1

Line 215: Table 6 – the information written in notes can be present in section 2.5, using equation.

Line 222: Table 7 – the information written in notes can be present in section 2.6, using equation.

Line 224: Figure 12 - suggestion insert scale on the images.

Line 235: Figure 13 - suggestion insert scale on the images.

Line 245: “melting point …“ – please insert references for each temperature.

Line 238 - 291: I agree with the description used in discussion because it describes the good evolution of the results obtained, however the discussion should be a little deeper, namely to use the results of the literature to make comparisons and whenever possible to explain the phenomena and mechanisms of improvement.

Line 289: “The possible cause …” - the conditional is not a good option, must be direct.

Conclusion it concise and summarize very well the principal remarks.

Author Response

Dear Reviewer,

Thank you very much for your valuable and substantive comments. Your opinion is very important for me. I accept all your suggestions. All your comments have been considered and corrected. In my opinion, the scientific and editorial value of my article has significantly improved. I hope you are satisfied with my work. I submitted the revised version of my article. The version includes corrections and suggestions from all reviewers.

Yours faithfully,

Artur Czupryński

Reviewer 2 Report

Dear Author,

The subject and the presented results are interesting, however I found some imperfections. The designations of the drawings do not comply with the educational requirements. Figures 1 to 4 should be modified, with appropriate arrangement of the figure descriptions.

The methodology should provide more technological details, such a processing parameters as well as technological conditions. Furthermore in the discussion section, in addition to the description of the obtained results, there should be an explanation of the observed phenomena in this study. I recommend improving the discussion section to provide further information for the reader.

Please correct row 138 of the manuscript: “(usunalem tutaj and)”

In addition, minor English corrections are necessary, so please verify the manuscript text.

Author Response

Dear Reviewer,

Thank you very much for your valuable and substantive comments. Your opinion is very important to me. I accept all your suggestions. I tried to include all the suggested corrections and improvements into the revised version.

I realise that the quality, porosity and adhesion of a coating are influenced by many factors, e.g. type of flame, particle velocity, process gas pressure, material surface preparation method, etc. Parameters of the subsonic flame spraying process are given in Table 3 and the description (lines 125-143). Optimal process parameters were determined based on preliminary tests. All samples were made at constant process parameters using a neutral flame. Your suggestion about measuring the size, temperature and speed of particles is very interesting, however I have not done such studies yet. It is an inspiration for further study on thermal spraying. I’m planning to conduct experiments with the FAST-IR infrared camera to capture the image of the jet spray.

However, at this moment I'm not able to extend my research due to lack of time -the revision process must be finished within 10 Days.

I hope that despite this the novelties presented in my article are sufficient to accept my manuscript for publication, because it is very important to me and crucial for the annual evaluation of a researcher at my University.

The main novelty of the presented study was the successful production of a composite coating reinforced with nanotubes, and characterized by unique properties such low friction coefficient and resistance to elevated temperature. Additionally, I pointed some potential new areas of application for such materials and coatings (lines 288-306).

I continue the study of thermal spraying, and I hope to publish more interesting results in future articles.

Once again I would like to thank you for you effort and comments, which improved the scientific and editorial value of my article. I hope you will be satisfied with all the revisions made.

The submitted revised version of my manuscript contains also improvements pointed by other Reviewers.

Yours faithfully,

Author

Reviewer 3 Report

Although the instruction has a long length, it has to improve its relevant to the research topic of flame spraying coating. The detail information of the experiment about spray is needed since it has play important role in determining the coating quality, such as air pressure, air flow rate, air temperature, distance between nozzle exit to surface, nozzle description, et al.. Could the authors measure the spray characteristics including spray pattern, droplet/particle size and velocity, temperature. It is quite important to the whole process and is also an interesting topic about spray. The description and discussion about the experimental results are too short. Please give some deep insight into the results. It should be nice if you can give your main innovation and contribution of this paper compared with the previous studies. Please give a careful check to the English writing. It can be further polished.

Author Response

(The authors gave the same response as above.)

Round 2

Reviewer 3 Report

It is suggested to add couples of important references relate to spray behavior in the introduction. "Experimental and theoretical studies on the droplet temperature behavior of R407C two-phase flashing spray[J]. International Journal of Heat and Mass Transfer, 2019, 136: 664-673." and "An experimental study on the spray and thermal characteristics of R134a two-phase flashing spray[J]. International journal of heat and mass transfer, 2012, 55(15-16): 4460-4468."

Other issues have been explained clearly.